# A Wearable Assistant Device for the Hearing Impaired to Recognize Emergency Vehicle Sirens with Edge Computing

**DOI:** 10.3390/s23177454

**Published:** 2023-08-27

**Authors:** Chiun-Li Chin, Chia-Chun Lin, Jing-Wen Wang, Wei-Cheng Chin, Yu-Hsiang Chen, Sheng-Wen Chang, Pei-Chen Huang, Xin Zhu, Yu-Lun Hsu, Shing-Hong Liu

**Affiliations:** 1Department of Medical Informatics, Chung Shan Medical University, Taichung 40201, Taiwan; ernestli@csmu.edu.tw (C.-L.C.); chiachunlin1223@gmail.com (C.-C.L.); s0958011@gm.csmu.edu.tw (J.-W.W.); s0958039@gm.csmu.edu.tw (W.-C.C.); s0958033@gm.csmu.edu.tw (Y.-H.C.); s0858048@gm.csmu.edu.tw (S.-W.C.); s0858009@gm.csmu.edu.tw (P.-C.H.); 2Division of Information Systems, School of Computer Science and Engineering, The University of Aizu, Aizu-Wakamatsu 965-8580, Fukushima, Japan; zhuxin@u-aizu.ac.jp; 3Bachelor’s Program of Sports and Health Promotion, Fo Guang University, Yilan 26247, Taiwan; ylhsu@gm.fgu.edu.tw; 4Department of Computer Science and Information Engineering, Chaoyang University of Technology, Taichung 41349, Taiwan

**Keywords:** edge computing, human vocalization, emergence vehicle siren, EfficientNet-based fuzzy rank-based ensemble model, hearing impairment

## Abstract

Wearable assistant devices play an important role in daily life for people with disabilities. Those who have hearing impairments may face dangers while walking or driving on the road. The major danger is their inability to hear warning sounds from cars or ambulances. Thus, the aim of this study is to develop a wearable assistant device with edge computing, allowing the hearing impaired to recognize the warning sounds from vehicles on the road. An EfficientNet-based, fuzzy rank-based ensemble model was proposed to classify seven audio sounds, and it was embedded in an Arduino Nano 33 BLE Sense development board. The audio files were obtained from the CREMA-D dataset and the Large-Scale Audio dataset of emergency vehicle sirens on the road, with a total number of 8756 files. The seven audio sounds included four vocalizations and three sirens. The audio signal was converted into a spectrogram by using the short-time Fourier transform for feature extraction. When one of the three sirens was detected, the wearable assistant device presented alarms by vibrating and displaying messages on the OLED panel. The performances of the EfficientNet-based, fuzzy rank-based ensemble model in offline computing achieved an accuracy of 97.1%, precision of 97.79%, sensitivity of 96.8%, and specificity of 97.04%. In edge computing, the results comprised an accuracy of 95.2%, precision of 93.2%, sensitivity of 95.3%, and specificity of 95.1%. Thus, the proposed wearable assistant device has the potential benefit of helping the hearing impaired to avoid traffic accidents.

## 1. Introduction

Hearing is an essential ability in daily life that aids in avoiding the risk of bodily injuries [1]. People with hearing impairments have reduced sensitivity to sound, leading to dangerous situations such as ignoring the warning sounds from vehicles. A study by Donmez and Gokkoca showed that elderly people are involved in 31.5% of traffic accidents in Turkey. Their hearing impairment was the main issue [2]. The reason for this is that elderly people who are afflicted with hearing impairments cannot perceive the warning sounds from vehicles. Tiwari and Ganveer proposed that 10.2% of those involved in traffic accidents are hearing impaired [3]. Therefore, the development of a wearable assistant device for recognizing the various warning sounds from ambulances and vehicles on the road could help the hearing impaired, reducing the risk of traffic accidents.

Recently, deep learning has been widely applied in voice recognition [4,5]. In 2021, Bonaventure et al. proposed the FSER architecture, which converts speech files into spectrograms and inputs them into a two-dimensional, convolutional neural network (2D CNN) for identification [6]. Its accuracy surpasses that of the one-dimensional convolutional neural network (1D CNN), as 2D CNN models can extract finer features from the spectrogram [7]. Kevin et al. aimed to build a more accurate sound classification model and proposed a two-stream neural network architecture that includes the EfficientNet-based model [8]. Lee et al. utilized preoperative and postoperative voice spectrograms as features to predict three-month postoperative vocal recovery [9]. This model could be widely applicable for transfer learning in sound classification. Lu et al. used the morphology of spectrograms as the input pattern to recognize speech using an EfficientNet model [10]. Padi et al. employed transfer learning to improve the accuracy of speech emotion recognition through spectrogram augmentation [7]. Additionally, Allamy and Koerich utilized a 1D CNN to classify music genres based on audio signals [11].

Ensemble learning is a powerful technique that involves the amalgamation of predictions from multiple classifiers to create a single classifier, resulting in notably enhanced accuracy compared to any individual classifier [12,13]. Research has demonstrated that an effective ensemble consists of individual classifiers with similar accuracies, yet with distributed errors across different aspects [14,15]. Essentially, ensemble learning encompasses two necessary characteristics: the generation of distinct individual classifiers, and their subsequent fusion. Two common strategies for generating individual classifiers include the heterogeneous type, which employs various learning algorithms, and the homogeneous type, which uses the same learning algorithm but requires different settings. Thus, Tan et al. proposed ensemble learning to classify human activities, combining a gated recurrent unit (GRU), a CNN stacked on the GRU, and a deep neural network [16]. Xie et al. proposed three DNN-based ensemble methods that fused a series of classifiers whose inputs are representations of intermediate layers [17]. Erdal et al. proposed a voting-based ensemble to improve identification results in tuberculosis classification, traditionally using a single CNN model [12,18]. This fusion method uses a voting algorithm to determine the output. However, its disadvantage is that it simply votes on the model’s output, only considering the number of predicted results and not the probability value of the predictions. Kavitha et al. proposed a weighted average-based ensemble to improve the accuracy of cell locations in cut electronic microscope images [19]. The disadvantage of this method is that when a large error occurs in the same prediction result, the weighted average result would be affected. Manna et al. proposed a fuzzy-based ensemble to improve the identification results of cervical cancer based on different CNN models. The output results of the CNN models, including InceptionV3, Xception, and DenseNet-169, were ensembled through fuzzy rank-based fusion [20]. The advantages of fuzzy rank-based ensembles include less computing time and memory consumption compared to fully connected layers.

The growth of the Internet of Everything (IoE) has led to great innovation in smart devices that connect to the internet, and in processing the necessary amounts of data. The innovation has aimed to resolve the problems of traditional cloud computing by decreasing burdensome bandwidth loads, increasing response speeds, and enhancing transmission security. To address these requirements, edge computing technologies have emerged as a promising solution [21,22]. Edge computing offers a more distributed and localized approach to data, allowing data to be processed in real time at the source. Hochst et al. proposed an edge artificial intelligence (AI) system to recognize bird species by their audio sounds, utilizing EfficientNet-B3 architecture based on an NVIDIA Jetson Nano board [23]. They demonstrated that the EfficientNet model could be efficiently implemented on an edge device. Rahman and Hossain developed an edge IoMT system using deep learning to detect various types of health-related COVID-19 symptoms based on a smartphone [24]. Nath et al. provided an overview of studies related to stress monitoring with edge computing, highlighting that computations performed using edge technology can reduce response time and are less vulnerable to external threats [25].

Based on the review of the above literature, the goal of this study is to develop a wearable assistant device with edge computing to help the hearing impaired to recognize the warning sounds from ambulances and vehicles on the road. An EfficientNet-based, fuzzy rank-based ensemble model was proposed to classify human vocalizations and warning sounds from vehicles. This model was embedded in an Arduino Nano 33 BLE Sense development board. The audio signals, including human vocalizations and the warning sounds of vehicles, were obtained from the CREMA-D dataset [26] and a Large-Scale Audio dataset of emergency vehicle sirens on the road [27], respectively. The categorization included seven types of audio sounds: neutral vocalization, anger vocalization, fear vocalization, happy vocalization, normal sound, car horn sound, siren sound, and ambulance siren sound. The spectrogram of the audio signal served as the feature. When one of the car horn, siren, or ambulance siren sounds was detected, the wearable assistant device presented alarms through a vibrator and displayed messages on the OLED panel. The results generated by edge computing were very close to those classified using offline computing. Moreover, we compared the performances between our proposed method and the iOS system, finding that our method outperformed the results of the iOS method. The contributions of this study included that the proposed EfficientNet-based, fuzzy rank-based ensemble model could be executed in the Arduino Nano 33 BLE Sense development board, and the performance of this model was better than the iOS system and the other deep learning models.

## 2. Materials and Methods

Figure 1 illustrates the architecture of the proposed method, including the data processing, training phase, and testing phase. The EfficientNet-based, fuzzy rank-based ensemble model is utilized to recognize human vocalizations and emergency vehicle sirens. During the training phase, this model is executed on a personal computer (PC). In contrast, during the testing phase, the proposed model is run on an Arduino Nano 33 BLE Sense development board. When the system detects specific sounds, such as car horns, sirens, or ambulance sirens, it alerts the user through the device’s vibrator and displays messages on an OLED panel. The audio signal is converted into a spectrogram to serve as the input pattern. The EfficientNet-based, fuzzy rank-based ensemble model includes three individual EfficientNet-based models and one fuzzy rank-based model. The EfficientNet-based models estimate the weight of each class. Then, the fuzzy rank-based model determines the winner of the classes. Once trained, the model is implemented in the wearable assistant device, drawing the user’s attention to warning sounds from vehicles.

### 2.1. Features

In this study, human vocalizations and emergency vehicle sirens on the road were classified. The human vocalizations, including neutral, anger, fear, and happy vocalizations, were sourced from the CREMA-D dataset, an emotional multimodal actor dataset [26]. A total of 5324 audio files were extracted from this dataset, and these files were augmented to create a total of 10,648 files. The emergency vehicle sirens on the road were acquired from the Large-Scale Audio dataset [27]. This dataset contains 3432 audio files featuring the car horn sound, siren sound, and ambulance siren sound. These files were augmented to a total of 6864 audio files using the RandAugment method [28]. The files were then divided into 80% for training, 10% for testing, and 10% for validation. The audio signals were segmented into 0.5 s intervals and transformed into spectrograms using the short-time Fourier transform. Each sample was represented by three spectrograms (1.5 s), which served as features for the classification of emergency vehicle sirens. Therefore, for the samples of emotional vocalizations, the numbers of training, testing, and validation samples were 25,555, 3195, and 3194, respectively. For the samples of emergency vehicle sirens, the numbers of training, testing, and validation samples were 16,474, 2058, and 2060, respectively.

### 2.2. EfficientNet-based Model

To recognize the human vocalizations and emergency vehicle sirens in real time with edge computing, the EfficientNet-based model was used to estimate the weights of all classes. The EfficientNet-based model is a deep learning model that has achieved top performance in image classification tasks and has demonstrated State-of-the-Art (SOTA) performance in the ImageNet image classification challenge [29]. The EfficientNet-based model builds on the base architectures of ResNet [30] and MobileNet [29,31] and leverages the compound scaling method [32] to achieve a balance between model size, computational efficiency, and accuracy. As a result, the EfficientNet-based model has become one of the most popular convolutional neural network models in current research. The structure of model is shown in Figure 2, which has one layer of Conv3 × 3, one layer of MBConv1, k3 × 3, six layers of MBConv6, k3 × 3, nine layers of MBConv6, k5 × 5, and one layer of full connection. The number of output layers is seven. The resolution and channel number of each layer are described in Table 1. Each row describes a stage i with $\hat{L}$^i^ layers, with input resolution {$\hat{H}$^i^, $\hat{W}$^i^} and output channels $\hat{C}$^i^. The hyperparameters of the EfficientNet-based model, as shown in Table 2, were used for all experiments. The optimizer is the Adam, learning rate is 1 × 10^−5^, batch size is 16, and the number of epochs is 1000. The sum of label smooth cross entropy loss function (*L_LSCE_*) [33] and focal cross entropy loss function (*L_FCE_*) [34] is defined as the total loss function (*L_T_*) to validate the performance of EfficientNet-based model.
(1)$$L_{LSCE}=\frac{-1}{N}\sum_{j=1}^{N}\sum_{i=1}^{M}P_{ji}log(f(x_{j}),$$
where *N* is the number of samples, *M* is the number of categories, f(∗) is the classifier, *x_j_* is the sample. When *x_j_* belongs to lth class, *P_ji_* = 1 − ε, and *P_ji_* = ε/*M* − 1 for the other classes. ε is 0.2.
(2)$$f\left(*\right)=\{\begin{array}{c} p,\;\;\;\;\;\;\mathrm{if}\;y=1\\ 1-p,\;\;\mathrm{otherwise} \end{array},$$
where *p* is the probability of the target.
(3)$$L_{FCE}=\frac{-1}{N}\sum_{j=1}^{N}\sum_{i=1}^{M}\alpha_{i}{\left(1-f(x_{j}\right)}^{\gamma }log(f(x_{j}),$$
where *α_i_* is the weight of loss function and *γ* is 2.
(4)$$L_{T}=L_{LSCE}+L_{FCE}$$

### 2.3. Fuzzy Rank-Based Model

The fuzzy rank-based model takes the results from the three EfficientNet-based models and calculates two fuzzy ranks by applying the exponential function and hyperbolic tangent function transformations, *R*_1_ and *R*_2_ [20].
(5)$$R_{1ik}=1-exp(-\frac{{\left(P_{ik}-1\right)}^{2}}{2}),\;k=1,\;2,\;3$$
(6)$$R_{2ik}=1-tanh(\frac{{\left(P_{ik}-1\right)}^{2}}{2}),\;k=1,\;2,\;3$$
where *P_ik_* is the estimating weight of ith category on kth EfficientNet-based model. *RS* is the fused rank score.
(7)$$RS_{ik}=R_{1ik}\times R_{2ik}$$

The confidence score of a particular category, *CS*, is the sum of *RS_sk_*.
(8)$$CS_{i}=\sum_{k=1}^{3}{RS}_{ik}$$

The category with the minimum confidence score is considered the winner.

### 2.4. Wearable Assistant Device

This system operates on the Arduino Nano 33 BLE Sense development board [35], which is linked to various hardware modules such as an OLED panel, vibrator, GPS positioning module, microphone, and relay module, as depicted in Figure 3. The relay module governs the power of the OLED panel to conserve energy, while the microphone captures the audio sound. When the wearable assistant device detects a car horn, siren, or ambulance siren, it activates the vibration module and displays a message on the OLED panel. Figure 4 shows a photo of the wearable assistant device being worn on the wrist. Its size is 3.9 cm × 3.1 cm × 2.5 cm.

### 2.5. Performances

The results of the tests are summarized in a confusion matrix, outlined in Table 3. This matrix illustrates the relationship between the actual and estimated classes in the test set, with each row corresponding to the actual classes and each column representing the estimated classes. According to the proposed method, a sample is classified as a true positive (TP) if the activity is correctly recognized, a false positive (FP) if the activity is incorrectly recognized, a true negative (TN) if the activity classification is correctly rejected, and a false negative (FN) if the activity classification is incorrectly rejected. To evaluate the performance of the proposed model, metrics such as accuracy, sensitivity, specificity, and precision are utilized, as described in Equations (9)–(12).
(9)$$\mathrm{Accuracy}=\frac{TP+TN}{TP+TN+FP+FN}$$
(10)$$\mathrm{Sensitivity}=\frac{TP}{TP+FN}$$
(11)$$\mathrm{Specificity}=\frac{TN}{TN+FP}$$
(12)$$\mathrm{Precision}=\frac{TP}{TP+FP}$$

## 3. Results

The proposed model underwent training, validation, and testing processes on a computer equipped with an 8-core CPU (Intel Xeon W-3223), 64 GB of RAM, a GPU (RTX 3090) with 24 GB of graphics memory, and 10,496 CUDA cores. PyTorch was the framework used for the implementation.

Table 4 presents the confusion matrix for the EfficientNet-based, fuzzy rank-based ensemble model in offline computing. The results clearly show that the study achieved an accuracy of 97.05%, a precision of 97.79%, a sensitivity of 96.8%, and a specificity of 97.04%.

In order to evaluate the robustness of the EfficientNet-based, fuzzy rank-based ensemble model in a noisy environment, we collected audio samples of road noise and superimposed the spectrogram of road noise over the testing samples. In this experiment, the number of testing samples is the same as the number used with the EfficientNet-based, fuzzy rank-based ensemble model in offline computing. Table 5 shows the confusion matrix of the EfficientNet-based, fuzzy rank-based ensemble model in offline computing under the noisy environment. It is evident that this study achieved an accuracy of 96.84%, precision of 96.17%, sensitivity of 96.13%, and specificity of 96.90%. We found that our proposed model exhibited high robustness. The accuracy, precision, sensitivity, and specificity only dropped 0.21%, 1.62%, 0.67%, and 0.14%, respectively.

The iOS system, developed by Apple Inc., Cupertino, CA, USA, is one of the most prevalent mobile operating systems, holding a substantial market share. Thanks to its vast user base, the iOS system has become highly popular. Starting with version 14, iOS has integrated sound recognition functionality, providing options to recognize various sounds such as alarms, car horns, and shouting, among others. To compare our proposed method with the iOS system, we utilized the sound recognition feature in iOS to classify emergency vehicle sirens [36]. The training, validation, and testing samples were the same as those used in the offline computing experiment with the EfficientNet-based, fuzzy rank-based ensemble model. Table 6 displays the confusion matrix for the iOS system in offline computing, showing an accuracy of 70.82%, precision of 76.67%, recall of 76.22%, and specificity of 71.02%. Comparing the results in Table 4 and Table 6, our proposed model outperformed the iOS system. The accuracy, precision, sensitivity, and specificity of our model exceeded those of the iOS system by 26.23%, 21.57%, 20.58%, and 26.02%, respectively.

In this study, we embedded the trained EfficientNet-based fuzzy rank-based ensemble model into the Arduino Nano 33 BLE Sense development board. The testing samples were played through the speaker of a PC. The board was recording the sounds and recognized the class of each sound. If the sound belonged to a car horn, siren, or ambulance siren, it would start the vibrator and send a message to the OLED. Then, we counted the numbers of all categories. The testing samples were the same as those used in the experiment on the EfficientNet-based, fuzzy rank-based ensemble model in offline computing. Table 7 shows the confusion matrix of the EfficientNet-based, fuzzy rank-based ensemble model in edge computing. The performances of edge computing achieved an accuracy of 95.22%, precision of 93.19%, sensitivity of 95.27%, and specificity of 95.09%. According to the results shown in Table 4 and Table 7, we find that the performances of edge computing are close to those of offline computing. Our proposed model exhibited the better performance. The accuracy, precision, sensitivity, and specificity only dropped 1.83%, 4.6%, 1.53%, and 1.95%, respectively. The testing video shows that the motorcycle horn and ambulance siren trigger the wearable assistant device, displaying the siren’s category [37].

## 4. Discussion

In ensemble learning, the voting and weight average methods are popular methods [38,39]. When a predicted class achieves the maximum voting number, the class is considered the winner. The main disadvantage of the voting method that it ignores the probabilities of true and false classes. It only focuses on the numbers of true classes. Although the weight average method could create a balance of true and false classes, this method would determine the wrong winner when the number of false classes is larger than the number of true classes. In this study, we proposed the fuzzy rank-based model to determine the winner amongst the classes. The model uses the two nonlinear functions, exponential function and hyperbolic tangent function, respectively, to estimate the ranks. The two functions are antisymmetric functions. Thus, they can inhibit the incorrect decisions caused by large amounts of false classes. Figure 5 shows the performances of the non-using ensemble (the EfficientNet-based model), voting and weight average methods (the EfficientNet-based voting ensemble model and the EfficientNet-based weight ensemble model), and the EfficientNet-based, fuzzy rank-based ensemble model. The results are: the accuracies of 83.31%, 87.23%, and 91.25%; the precisions of 82.87%, 86.82%, and 91.73%; the sensitivities of 83.01%, 87.11%, and 90.96%; and the specificities of 83.33%, 86.74%, and 91.17%. The performances of the EfficientNet-based, fuzzy rank-based ensemble model are the best, with an accuracy of 97.05%, precision of 97.79%, sensitivity of 96.8%, and specificity of 97.04%.

When compared to the iOS system, our method demonstrated significantly higher accuracy, outperforming it by a remarkable margin of 26.23%. This performance could be considered the main contribution of this study. The advantage of the EfficientNet-based, fuzzy rank-based ensemble model is that it not only recognized the warning sounds, but also identified the emotions within human vocalizations. The iOS system lacks the ability to recognize emotion. This reason indicates that our method is better than the iOS system.

Table 8 presents a comparative analysis of our proposed method against other studies that utilized the CREMA-D dataset [26]. Previous studies [40,41,42,43] only classified four sounds, while our study classified seven. As shown, the proposed EfficientNet-based, fuzzy rank-based ensemble model achieved an accuracy of 97.05%, which ranks the best result within these literatures. For the Large-Scale Audio dataset of emergency vehicle sirens on the road [16], a previous study showed that the best result for the classification of four sounds was an accuracy of 97%. This result was very close to that of our study. However, we emphasized that our approach classified seven sounds, broadening the scope of sound recognition.

In this study, we used two datasets, the CREMA-D dataset and the Large-Scale Audio dataset to evaluate the performance of the EfficientNet-based, fuzzy rank-based ensemble model. The number of each category is very different. The samples’ numbers of neutral vocalizations, anger vocalizations, fear vocalizations, happy vocalizations, car horn sounds, siren sounds, and ambulance sirens sound were 7921, 7834, 7621, 8568, 7128, 7484, and 5979, respectively. Moreover, the target output used one-hot encoding. If the model uses general cross entropy loss, the model may have poor training results due to a sample imbalance. The label smooth cross entropy loss function and focal loss function could overcome these problems [33,34]. Thus, we used the sum of two loss functions to validate the performance of the proposed model to avoid overfitting and local minimum problems in the training phase.

## 5. Conclusions

We proposed a wearable assistant device to help the hearing impaired recognize the warning sounds from vehicles on the road. By employing edge computing in the Arduino Nano 33 BLE Sense development board, we executed the EfficientNet-based, fuzzy rank-based ensemble model. To evaluate the performance of this model, we used the CREMA-D dataset and the Large-Scale Audio dataset for training and testing. In offline computing, the accuracy reached 97.05%, while in edge computing, it also achieved 95.22% accuracy. Because the audio signals were obtained from the open datasets, we did not explore the performance of the wearable assistant device under different types of audio signals and background noises. These problems will directly affect the response time of the user when faced with an approaching vehicle. In the future, we will explore the sensitivity and robustness of the wearable assistant device and increase the response time of the user.

## Figures and Tables

**Figure 1 sensors-23-07454-f001:**
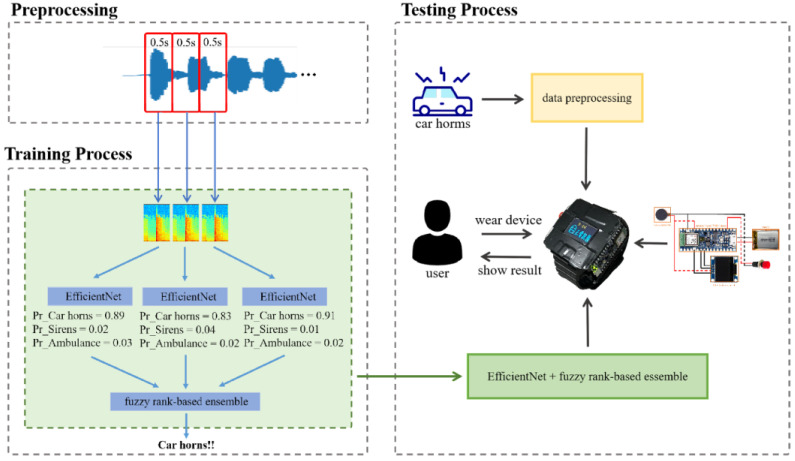
The architecture of proposed wearable assistant system allowing the hearing impaired to recognize warning sounds from vehicles.

**Figure 2 sensors-23-07454-f002:**
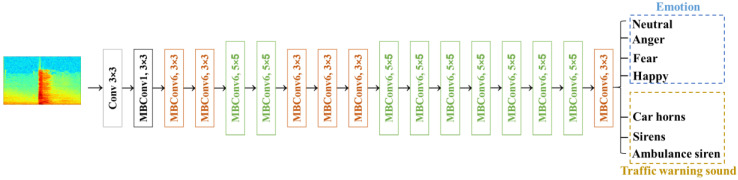
The architecture of EfficientNet-based model.

**Figure 3 sensors-23-07454-f003:**
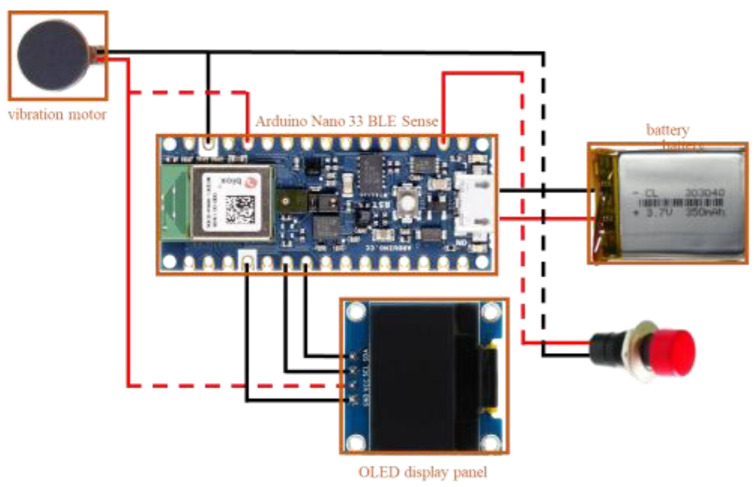
The Arduino Nano 33 BLE Sense development board.

**Figure 4 sensors-23-07454-f004:**
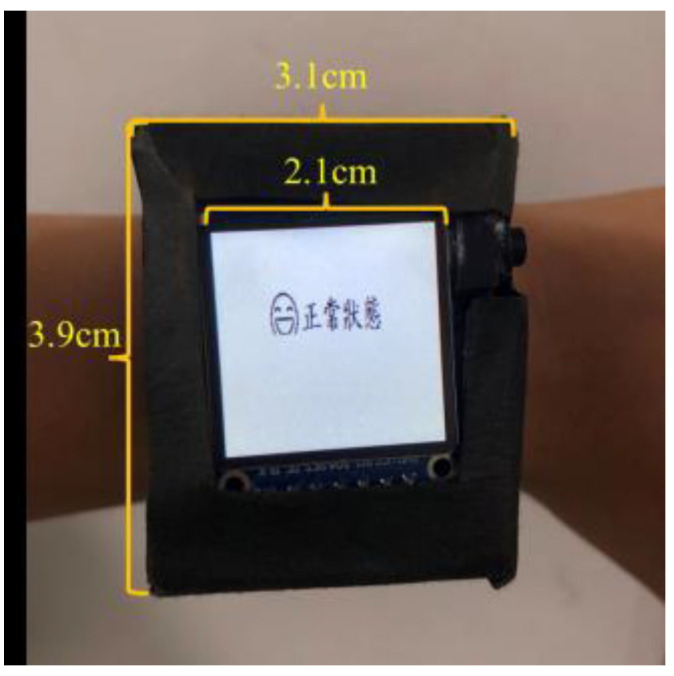
The photo of the wearable assistant device being worn on the wrist, the size of which is 3.9 cm × 3.1 cm × 2.5 cm.

**Figure 5 sensors-23-07454-f005:**
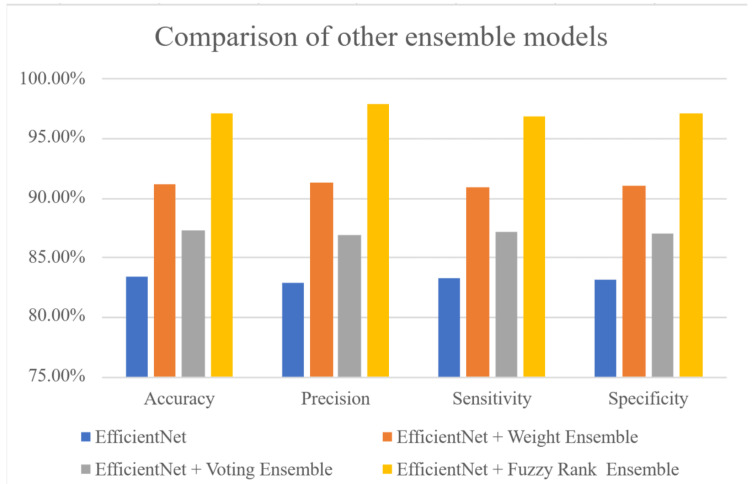
The comparison of other ensemble models. The EfficientNet-based, fuzzy rank-based ensemble model has an accuracy of 97.05%, precision of 97.79%, sensitivity of 96.8%, and specificity of 97.04%.

**Table 1 sensors-23-07454-t001:** EfficientNet-based model, the stage i with ${\hat{L}}_{i}$ layers, with input resolution $\langle {\hat{H}}_{I},\;{\hat{W}}_{i}\rangle$ and output channels ${\hat{C}}_{i}$.

stage i	Operator ${\hat{F}}_{i}$.	Resolution ${\hat{H}}_{I}\times {\hat{W}}_{i}$	#Channels ${\hat{C}}_{i}$	#Layers ${\hat{L}}_{i}$
1	Conv3 × 3	224 × 192	32	1
2	MBConv1, k3 × 3	112 × 96	16	1
3	MBConv6, k3 × 3	112 × 96	24	2
4	MBConv6, k5 × 5	56 × 48	40	2
5	MBConv6, k3 × 3	28 × 24	80	3
6	MBConv6, k5 × 5	28 × 24	112	3
7	MBConv6, k5 × 5	14 × 12	192	4
8	MBConv6, k3 × 3	7 × 6	320	1
9	Conv1 × 1 & Flatten & FC	7 × 6	1280	1

**Table 2 sensors-23-07454-t002:** Hyperparameters for training.

Hyperparameter	Selected Value
Loss function	$L_{Total}=L_{LSCE}+L_{FL}$
Optimizer	Adam
Learning rate	1 $\times {\;10}^{-5}$
Batch size	16
Epoch	1000

**Table 3 sensors-23-07454-t003:** Confusion matrix.

		Estimated Class	
Actual Class		Positive	Negative
	Positive	*TP* (True Positive)	*FN* (False Negative)
	Negative	*FP* (False Positive)	*TN* (True Negative)

**Table 4 sensors-23-07454-t004:** The confusion matrix of EfficientNet-based, fuzzy rank-based ensemble model in offline computing.

	Neutral	Anger	Fear	Happy	Car Horns	Sirens	Ambulance Siren
**Neutral**	2590	10	5	4	0	0	0
**Anger**	10	2588	14	5	5	0	0
**Fear**	6	11	2594	3	0	2	0
**Happy**	8	6	2	2598	0	0	0
**Car Horns**	0	0	0	0	1094	17	25
**Sirens**	0	0	0	0	13	1105	35
**Ambulance Siren**	0	0	1	0	26	21	1096

**Table 5 sensors-23-07454-t005:** The confusion matrix of the EfficientNet-based, fuzzy rank-based ensemble model in offline computing under the noisy environment. The testing samples were superimposed over the road noise.

	Neutral	Anger	Fear	Happy	Car Horns	Sirens	Ambulance Siren
**Neutral**	2538	20	10	12	0	0	0
**Anger**	35	2549	33	20	0	0	0
**Fear**	24	27	2568	7	0	0	0
**Happy**	17	19	3	2571	0	0	0
**Car Horns**	0	0	0	0	1079	66	46
**Sirens**	0	0	0	0	18	1067	28
**Ambulance Siren**	0	0	0	0	39	12	1082

**Table 6 sensors-23-07454-t006:** The confusion matrix of iOS system in the offline computing. All samples are the same as in Table 4.

	Neutral	Anger	Fear	Happy	Car Horns	Sirens	Ambulance Siren
**Neutral**	1927	215	192	794	0	0	0
**Anger**	143	1589	776	118	0	0	0
**Fear**	230	743	1521	101	0	0	0
**Happy**	314	68	127	1597	0	0	0
**Car Horns**	0	0	0	0	1065	23	16
**Sirens**	0	0	0	0	26	1045	43
**Ambulance Siren**	0	0	1	0	47	77	1097

**Table 7 sensors-23-07454-t007:** The confusion matrix of the EfficientNet-based fuzzy rank-based ensemble model in edge computing. The testing samples were the same as in the experiment involving offline computing.

	Neutral	Anger	Fear	Happy	Car Horns	Sirens	Ambulance Siren
**Neutral**	1272	14	12	11	8	12	10
**Anger**	12	1274	13	23	10	8	12
**Fear**	23	8	1268	12	5	10	11
**Happy**	9	10	4	1267	7	3	5
**Car Horns**	15	20	10	8	1084	14	6
**Sirens**	7	2	18	9	13	1079	7
**Ambulance Siren**	10	12	9	14	9	5	1095

**Table 8 sensors-23-07454-t008:** Comparative results of various methods using the CREMA-D dataset.

Ref.	Classification Method	*F*_1_-Score (%)	Accuracy (%)
[40]	ResNet18	NA	57.42%
[41]	CNN-LSTM	79.23%	78.52%
[42]	Metric Learning-Based Multimodal	NA	65.01%
[43]	Triplet Loss-based modal	NA	58.72%
NA	EfficientNet-based fuzzy rank-based ensemble model	NA	97.05%.

## Data Availability

https://paperswithcode.com/dataset/crema-d, and https://github.com/tabarkarajab/Large-Scale-Audio-dataset (accessed on 6 January 2023).

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
