# Peer review of "A Wearable Assistant Device for the Hearing Impaired to Recognize Emergency Vehicle Sirens with Edge Computing"

_sensors, 2023, doi:10.3390/s23177454_

Round 1

Reviewer 1 Report

This study develops a wearable assistant device for the hearing impaired to recognize emergency vehicle sirens on the road using edge computing. By employing an edge computing method in the Arduino Nano 33 BLE Sense development board, the authors executed the Efficient Net-based fuzzy rank-based ensemble model. To evaluate the performance of this model, the authors also used the CREMA-D dataset and the Large-Scale Audio dataset for training and testing. This paper demonstrates that the Efficient Net-based fuzzy rank-based ensemble model has the potential to be applied in edge computing for other image classifications. Considering a detailed study has been carried out, I have only the below comments addressing paper writing, and once these are addressed the work is recommended for publication.

(1)   There is a great deal of repetition in Figures 1 and 2, and it is recommended that Figure 1 be redrawn, or replaced.

(2)   Add some experiment videos or examples of applications as supporting material to enhance the completeness and interest of this article.

(3)  Correction of keywords part (“Efficient Net-based 33 fuzzy rank-based ensemble model”) and make it a more concise presentation.

(4)   Expand the concluding section to include a future outlook, especially for applications.

Author Response

To Reviewer #1:

Thank the first reviewer for his/her valuable comments that make better this manuscript. The texts in this revised manuscript have been corrected/ modified by red words. It is our sincere hope that this revision will enhance readability and strengthen of the manuscript to satisfy the requirements of this prestigious journal.

Comments and Suggestions for Authors

This study develops a wearable assistant device for the hearing impaired to recognize emergency vehicle sirens on the road using edge computing. By employing an edge computing method in the Arduino Nano 33 BLE Sense development board, the authors executed the Efficient Net-based fuzzy rank-based ensemble model. To evaluate the performance of this model, the authors also used the CREMA-D dataset and the Large-Scale Audio dataset for training and testing. This paper demonstrates that the Efficient Net-based fuzzy rank-based ensemble model has the potential to be applied in edge computing for other image classifications. Considering a detailed study has been carried out, I have only the below comments addressing paper writing, and once these are addressed the work is recommended for publication.

  1. There is a great deal of repetition in Figures 1 and 2, and it is recommended that Figure 1 be redrawn, or replaced.

ANS: Many thanks for reviewer’s comment. We deleted Fig. 2 and sector 2.2, and modified the contexts of Sector 2. Materials and Methods to describe the EfficientNet-based fuzzy rank-based ensemble model more clear. In Sector 2.2 EfficientNet-based Model, we only describe the detail structure of the EfficientNet-based model.

Line 122-135

  1. Materials and Methods

Figure 1 illustrates the architecture of the proposed method, including the data processing, training phase, and testing phase. The EfficientNet-based fuzzy rank-based ensemble model is utilized to recognize human vocalizations and emergency vehicle sirens. During the training phase, this model is executed on a personal computer (PC). In contrast, during the testing phase, the proposed model is run on an Arduino Nano 33 BLE Sense development board. When the system detects specific sounds, such as car horns, sirens, or ambulance sirens, it alerts the user through the device's vibrator and displays messages on an OLED panel. The audio signal is converted into a spectrogram to serve as the input pattern. The EfficientNet-based fuzzy rank-based ensemble model includes three individual EfficientNet-based models and one fuzzy rank-based model. The EfficientNet-based models estimates the weight of each class. Then, the fuzzy rank-based model determines the winner of classes. Once trained, the model is implemented in the wearable device, promoting the user's attention for warning sounds from vehicles.

  1. Add some experiment videos or examples of applications as supporting material to enhance the completeness and interest of this article.

ANS: Many thanks for reviewer’s comment. We added one photo of the wearable assistant device wore on the wrist and its size in Sector 2.4 Wearable Assistant Device. Moreover, in the fifth paragraph of Sector 3. Results, we add a sentence to show the performance of testing video.

Line: 204-212

2.4. Wearable Assistant Device

This system operates on the Arduino Nano 33 BLE Sense development board [35], linked to various hardware modules such as an OLED panel, vibrator, GPS positioning module, microphone, and relay module, as depicted in Fig. 3. The relay module governs the power of the OLED panel to conserve energy, while the microphone captures the audio sound. When the wearable assistant device detects a car horn, siren, or ambulance siren, it activates the vibration module and displays a message on the OLED panel. Figure 4 shows the photo of the wearable assistant device wore on the wrist. Its size is 3.9 cm × 3.1 cm × 2.5 cm.

Figure 4. The photo of wearable assistant device wore on the wrist, which size is 3.9 cm × 3.1 cm × 2.5 cm.

Line 268-282:

In this study, we imbedded the trained EfficientNet-based fuzzy rank-based ensemble model to the Arduino Nano 33 BLE Sense development board. The testing samples were played by the speaker of PC. This board was recording the sound and recognize the class of this sound. If the sound belongs to the car horn, siren, and ambulance siren, it would start the vibrator and send the message to the OLED. Then, we counted the numbers of all categories. The testing samples were same to the experiment of EfficientNet-based fuzzy rank-based ensemble model in the offline computing. Table 7 shows the confusion matrix of EfficientNet-based fuzzy rank-based ensemble model in the edge computing. The performances of edge computing achieved an accuracy of 95.22%, precision of 93.19%, sensitivity of 95.27%, and specificity of 95.09%. According to the results shown in Table 4 and 7, we find that the performances of edge computing are close to the offline computing. our proposed model exhibited the better performance. The accuracy, precision, sensitivity, and specificity only drop 1.83%, 4.6%, 1.53%, and 1.95%, respectively. The testing video show that the motorcycle horn and ambulance siren trigger the wearable assistant device displaying the siren’s category [37].

  1. https://drive.google.com/file/d/1NO1FpZ4LxTDhiH1B0ZW7RQoiTlb0gQlF/view?usp=drive_link

  1. Correction of keywords part (“Efficient Net-based fuzzy rank-based ensemble model”) and make it a more concise presentation.

ANS: Many thanks for reviewer’s comment. We corrected the description of EfficientNet-based fuzzy rank-based ensemble model that includes three EfficientNet-based models and one fuzzy rank-based model.

  1. Expand the concluding section to include a future outlook, especially for applications.

ANS: Many thanks for reviewer’s comment. In the Sector 5. Conclusion, we added some sentences to describe the limitation of this study and the future work.

Line 335-346:

  1. Conclusions

We proposed a wearable assistant device to help the hearing-impaired recognize the warning sounds from vehicles on the road. By employing the edge computing in the Arduino Nano 33 BLE Sense development board, we executed the EfficientNet-based fuzzy rank-based ensemble model. To evaluate the performance of this model, we used the CREMA-D dataset and Large Scale Audio dataset for training and testing. In offline computing, the accuracy reached 97.05%, while in edge computing, it also achieved 95.22% of accuracy. Because the audio signals were obtained from the open datasets, we do not explore the performance of the wearable assistant device under the different powers of audio signals and background noises. These problems will directly affect the response time of user for the approaching vehicle. In the future, we will explore the sensitivity and robust of the wearable assistant device and increase the response time of user.

Reviewer 2 Report

This study investigates a wearable assistant device for the hearing impaired to recognize emergency vehicle sirens on the road using edge computing, and an efficientnet-based fuzzy rank-based ensemble model was proposed to classify seven audio sounds, including human vocalizations and emergency vehicle sirens. Upon a detailed review, the reviewer believes that the manuscript is far from the standard that is used in this journal and cannot recommend it for publication. The authors can review the below for reasons behind this decision. More trial information and results should be given, apparently, currently lacking.

Although this manuscript contains an in-depth discussion, the limited experimental content prevents it from meeting publication requirements.

The current view of the manuscript is so weak that the reviewers cannot get enough useful information.

The description in the introduction is incoherent and needs further modification. Meanwhile, the novelty of the article should be highlighted at the end of the introduction.

Please check the grammar throughout the document.

The authors are encouraged to add more critical analysis of the data that they have provided. In particular, standard deviation and hypothesis testing are highly suggested to see the statistical significance of the test results.

Good

Author Response

To Reviewer #2:

Thank the second reviewer for his/her valuable comments that make better this manuscript. The texts in this revised manuscript have been corrected/ modified by red words. It is our sincere hope that this revision will enhance readability and strengthen of the manuscript to satisfy the requirements of this prestigious journal.

Comments and Suggestions for Authors

Comments and Suggestions for Authors

This study investigates a wearable assistant device for the hearing impaired to recognize emergency vehicle sirens on the road using edge computing, and an efficientnet-based fuzzy rank-based ensemble model was proposed to classify seven audio sounds, including human vocalizations and emergency vehicle sirens. Upon a detailed review, the reviewer believes that the manuscript is far from the standard that is used in this journal and cannot recommend it for publication. The authors can review the below for reasons behind this decision. More trial information and results should be given, apparently, currently lacking.

ANS: Many thanks for reviewer’s comment. I offer an apology for the contents in Sector 1. Introduction which have many incoherent contexts and confusion descriptions. Thus, we modified the contexts of Introduction sector to let readers have the interesting and easily understand its potential knowledges of edge computing. 

  1. Although this manuscript contains an in-depth discussion, the limited experimental content prevents it from meeting publication requirements.

ANS: Many thanks for reviewer’s comment. The major contribution of this study is that the edge computing for the classification of the warring sounds from the ambulance and vehicles was realized in an Arduino Nano 33 BLE Sense development board which size is 3.9 cm × 3.1 cm. The challenge of edge computing is to explore the balance of model size and performance. Thus, we proposed an EfficientNet-based fuzzy rank-based ensemble model to recognize the warning sounds. In order to show the benefits of the EfficientNet-based fuzzy rank-based ensemble model, we used the open datasets, the CREMA-D dataset and Large Scale Audio dataset of emergency vehicle sirens on the road, to verify the performances of this model in the offline computing and online computing. Moreover, in the benchmark trial, we also used the iOS system to recognize these warning sounds. We found that our proposed model had the better performance than the iOS system. In the tolerance trial, the testing samples were added the noise on the road. The results of all trials were described by the confusion matrices in Sector 3. Results. We added one photo of the wearable assistant device wore on the wrist and its size in Sector 2.4 Wearable Assistant Device. Moreover, in the fifth paragraph of Sector 3. Results, we add a sentence to show the performance of testing video.

Line 36-121:

  1. Introduction

The hearing is an essential ability of daily life for avoiding the risk of body injuries [1]. People with hearing impairment would reduce the sensitivity to sound recognition, leading to the inconvenience, like as ignoring the warning sounds from vehicles. The study of Donmez and Gokkoca showed that elders of 31.5% happened the traffic accidents in Turkey. The hearing impairment was the one issue [2]. The reason is that elders with the hearing impairment cannot get the warn from the vehicles. Tiwari and Ganveer proposed that the traffic accident of 10.2% was the hearing impaired [3]. Therefore, the development of a wearable assistant device for recognizing the various warning sounds from ambulances and vehicles on the road could help the hearing impaired reducing the risk of traffic accidents.

Recently, deep learning has been widely applied in voice recognition [4,5]. In 2021, Bonaventure et al. proposed the FSER architecture, which converts speech files into spectrograms and inputs them into a two-dimensional convolutional neural network (2D CNN) for identification [6]. Its accuracy surpasses that of the one-dimensional convolutional neural network (1D CNN), as 2D CNN models can extract finer features from the spectrogram [7]. Kevin et al. aimed to build a more accurate sound classification model and proposed a two-stream neural network architecture that includes the EfficientNet-based model [8]. Lee et al. utilized preoperative and postoperative voice spectrograms as features to predict three-month postoperative vocal recovery [9]. This model could be widely applied for transfer learning in sound classification. Lu et al. used the morphology of spectrograms as the input pattern to recognize speech using EfficientNet model [10]. Padi et al. employed transfer learning to improve the accuracy of speech emotion recognition through spectrogram augmentation [7]. Additionally, Allamy and Koerich utilized a 1D CNN to classify music genres based on audio signals [11].

Ensemble learning is a powerful technique that involves the amalgamation of predictions from multiple classifiers to create a single classifier, resulting in notably enhanced accuracy compared to any individual classifier [12,13]. Research has demonstrated that an effective ensemble consists of individual classifiers with similar accuracies, yet with distributed errors across different aspects [14,15]. Essentially, ensemble learning encompasses two necessary characteristics: the generation of distinct individual classifiers and their subsequent fusion. Two common strategies for generating individual classifiers include the heterogeneous type, which employs various learning algorithms, and the homogeneous type, which uses the same learning algorithm but requires different settings. Thus, Tan et al. proposed ensemble learning to classify human activities, combining a gated recurrent unit (GRU), a CNN stacked on the GRU, and a deep neural network [16]. Xie et al. proposed three DNN-based ensemble methods that fused a series of classifiers whose inputs are representations of intermediate layers [17]. Erdal et al. proposed a voting-based ensemble to improve identification results in tuberculosis classification, traditionally using a single CNN model [18,12]. This fusion method used a voting algorithm to determine the output. However, its disadvantage is that it simply votes on the model's output, only considering the number of predicted results and not the probability value of those predictions. Kavitha et al. proposed a weighted average-based ensemble to improve the accuracy of cell locations in cut electronic microscope images [19]. The disadvantage of this method is that when a large error occurs in the same prediction result, the weighted average result would be affected. Manna et al. proposed a fuzzy-based ensemble to improve the identification results of cervical cancer based on different CNN models. The output results of CNN models, including InceptionV3, Xception, and DenseNet-169, were ensembled through fuzzy rank-based fusion [20]. The advantages of fuzzy rank-based ensemble include less computing time and memory consumption compared to fully connected layers.

The growth of the Internet of Everything (IoE) has led to a great innovation in smart devices connecting to the internet, and processing the amount of data. The innovation has to improve the problems of traditional cloud computing, like as decreasing the burdensome bandwidth loads, increasing the response speeds, and enhancing the transmitting security. To address these requirements, the edge computing technologies have been emerged as a promising solution [21,22]. Edge computing offers a more distributed and localized approach of data, allowing data to be processed in real time at the source. Hochst et al. proposed an edge artificial intelligence (AI) system to recognize bird species by their audio sounds, utilizing the EfficientNet-B3 architecture based on an NVIDIA Jetson Nano board [23]. They demonstrated that the EfficientNet model could be efficiently implemented on an edge device. Rahman and Hossain developed an edge IoMT system using deep learning to detect various types of health-related COVID-19 symptoms based on a smartphone [24]. Nath et al. provided an overview of studies related to stress monitoring with edge computing, highlighting that computations performed at the edge can reduce response time and are less vulnerable to external threats [25].

Based on the review of the above literature, the goal of this study is to develop a wearable assistant device with the edge computing for helping the hearing impaired to recognize the warning sounds from the ambulance and vehicles on the road. An EfficientNet-based fuzzy rank-based ensemble model was proposed to classify human vocalizations and warning sounds of vehicles. This model was embedded in an Arduino Nano 33 BLE Sense development board. The audio signals, including human vocalizations and warning sounds of vehicles, were obtained from the CREMA-D dataset [26] and Large Scale Audio dataset of emergency vehicle sirens on the road [27], respectively. The categorization had seven types of audio sounds: neutral vocalization, anger vocalization, fear vocalization, happy vocalization, normal sound, car horn sound, siren sound, and ambulance siren sound. The spectrogram of the audio signal served as the feature. When one of the car horn, siren, or ambulance siren sounds was detected, the wearable assistant device presented alarms through a vibrator and displayed messages on the OLED panel. The results in edge computing were very close to those classified by offline computing. Moreover, we compared the performances between our proposed method and the iOS system, finding that our method outperformed the results of the iOS method. The contributions of this study included that the proposed EfficientNet-based fuzzy rank-based ensemble model could be executed in the Arduino Nano 33 BLE Sense development board, and the performance of this model was better than the iOS system and the other deep learning models.    

Line: 204-212

2.4. Wearable Assistant Device

This system operates on the Arduino Nano 33 BLE Sense development board [35], linked to various hardware modules such as an OLED panel, vibrator, GPS positioning module, microphone, and relay module, as depicted in Fig. 3. The relay module governs the power of the OLED panel to conserve energy, while the microphone captures the audio sound. When the wearable assistant device detects a car horn, siren, or ambulance siren, it activates the vibration module and displays a message on the OLED panel. Figure 4 shows the photo of the wearable assistant device wore on the wrist. Its size is 3.9 cm × 3.1 cm × 2.5 cm.

Figure 4. The photo of wearable assistant device wore on the wrist, which size is 3.9 cm × 3.1 cm × 2.5 cm.

Line 268-282:

In this study, we imbedded the trained EfficientNet-based fuzzy rank-based ensemble model to the Arduino Nano 33 BLE Sense development board. The testing samples were played by the speaker of PC. This board was recording the sound and recognize the class of this sound. If the sound belongs to the car horn, siren, and ambulance siren, it would start the vibrator and send the message to the OLED. Then, we counted the numbers of all categories. The testing samples were same to the experiment of EfficientNet-based fuzzy rank-based ensemble model in the offline computing. Table 7 shows the confusion matrix of EfficientNet-based fuzzy rank-based ensemble model in the edge computing. The performances of edge computing achieved an accuracy of 95.22%, precision of 93.19%, sensitivity of 95.27%, and specificity of 95.09%. According to the results shown in Table 4 and 7, we find that the performances of edge computing are close to the offline computing. our proposed model exhibited the better performance. The accuracy, precision, sensitivity, and specificity only drop 1.83%, 4.6%, 1.53%, and 1.95%, respectively. The testing video show that the motorcycle horn and ambulance siren trigger the wearable assistant device displaying the siren’s category [37].

  1. https://drive.google.com/file/d/1NO1FpZ4LxTDhiH1B0ZW7RQoiTlb0gQlF/view?usp=drive_link

  1. The current view of the manuscript is so weak that the reviewers cannot get enough useful information. The description in the introduction is incoherent and needs further modification. Meanwhile, the novelty of the article should be highlighted at the end of the introduction.

ANS: Many thanks for reviewer’s comment. We modified the contents of Sector 1. Introduction to let more coherent. Moreover, we also added some sentences to highlight the contribution of this study.

  1. Please check the grammar throughout the document.

ANS: Many thanks for reviewer’s comment. We have carefully checked the grammar of this manuscript again.

  1. The authors are encouraged to add more critical analysis of the data that they have provided. In particular, standard deviation and hypothesis testing are highly suggested to see the statistical significance of the test results.

ANS: Many thanks for reviewer’s comment. In this study, the aim is to develop the wearable assistant device with the edge computing that is a deep learning technique. The confusion matrix is the most useful analysis method according to the text book of pattern recognition [1]. We used the parameters of accuracy, precision, sensitivity, and specificity to evaluate the performances of our proposed method. According to results of the four parameters, the EfficientNet-based fuzzy rank-based ensemble model showed the best performance. We do not use the t-test to analyze the significant difference between the results of two methods.

  1. Christopher M. Bishop, Pattern Recognition and Machine Learning, ISBN-10: 0-387-31073-8, 2006 Springer Science+Business Media, LLC.

Reviewer 3 Report

This work is very relevant due the increase of people with the  Hearing Impaired ,Due that,one of my sugesstion to improve is to include a refrence of the amount of people with this problem at least in the last 5 years.

Other recommendation if is possible to include the sizes and weight of the wearable artefact , 

Author Response

To Reviewer #3:

Thank the third reviewer for his/her valuable comments that make better this manuscript. The texts in this revised manuscript have been corrected/ modified by red words. It is our sincere hope that this revision will enhance readability and strengthen of the manuscript to satisfy the requirements of this prestigious journal.

Comments and Suggestions for Authors

  1. This work is very relevant due the increase of people with the Hearing Impaired, Due that, one of my suggestion to improve is to include a reference of the amount of people with this problem at least in the last 5 years.

ANS: Many thanks for reviewer’s comment. We added two references about the traffic accident and hearing impairment in the first paragraph of Sector 1 Introduction.

Line 37-46:

The hearing is an essential ability of daily life for avoiding the risk of body injuries [1]. People with hearing impairment would reduce the sensitivity to sound recognition, leading to the inconvenience, like as ignoring the warning sounds from vehicles. The study of Donmez and Gokkoca showed that elders of 31.5% happened the traffic accidents in Turkey. The hearing impairment was the one issue [2]. The reason is that elders with the hearing impairment cannot get the warn from the vehicles. Tiwari and Ganveer proposed that the traffic accident of 10.2% was the hearing impaired [3]. Therefore, the development of a wearable assistant device for recognizing the various warning sounds from ambulances and vehicles on the road could help the hearing impaired reducing the risk of traffic accidents.

  1. Donmez, L.; Gokkoca, Z. Accident profile of older people in Antalya city center, Turkey. Archives of Gerontology and Geriatrics 2003, 37(2), 99-108.
  2. Tiwari, R. R.; Ganveer, G. B. A study on human risk factors in non-fatal road traffic accidents at Nagpur. Indian Journal of Public Health 2008, 52(4), 197-199.

  1. Other recommendation if is possible to include the sizes and weight of the wearable artefact.

ANS: Many thanks for reviewer’s comment. We added one photo of the wearable assistant device wore on the wrist and its size in Sector 2.4 Wearable Assistant Device. Moreover, in the fifth paragraph of Sector 3. Results, we add a sentence to show the performance of testing video.

Line: 204-212

2.4. Wearable Assistant Device

This system operates on the Arduino Nano 33 BLE Sense development board [35], linked to various hardware modules such as an OLED panel, vibrator, GPS positioning module, microphone, and relay module, as depicted in Fig. 3. The relay module governs the power of the OLED panel to conserve energy, while the microphone captures the audio sound. When the wearable assistant device detects a car horn, siren, or ambulance siren, it activates the vibration module and displays a message on the OLED panel. Figure 4 shows the photo of the wearable assistant device wore on the wrist. Its size is 3.9 cm × 3.1 cm × 2.5 cm.

Figure 4. The photo of wearable assistant device wore on the wrist, which size is 3.9 cm × 3.1 cm × 2.5 cm.

Line 268-282:

In this study, we imbedded the trained EfficientNet-based fuzzy rank-based ensemble model to the Arduino Nano 33 BLE Sense development board. The testing samples were played by the speaker of PC. This board was recording the sound and recognize the class of this sound. If the sound belongs to the car horn, siren, and ambulance siren, it would start the vibrator and send the message to the OLED. Then, we counted the numbers of all categories. The testing samples were same to the experiment of EfficientNet-based fuzzy rank-based ensemble model in the offline computing. Table 7 shows the confusion matrix of EfficientNet-based fuzzy rank-based ensemble model in the edge computing. The performances of edge computing achieved an accuracy of 95.22%, precision of 93.19%, sensitivity of 95.27%, and specificity of 95.09%. According to the results shown in Table 4 and 7, we find that the performances of edge computing are close to the offline computing. our proposed model exhibited the better performance. The accuracy, precision, sensitivity, and specificity only drop 1.83%, 4.6%, 1.53%, and 1.95%, respectively. The testing video show that the motorcycle horn and ambulance siren trigger the wearable assistant device displaying the siren’s category [37].

  1. https://drive.google.com/file/d/1NO1FpZ4LxTDhiH1B0ZW7RQoiTlb0gQlF/view?usp=drive_link

Round 2

Reviewer 1 Report

Accept in present form

Reviewer 2 Report

The authors have revised the manuscript.